# An SVM Based Weight Scheme for Improving Kinematic GNSS Positioning Accuracy with Low-Cost GNSS Receiver in Urban Environments

**DOI:** 10.3390/s20247265

**Published:** 2020-12-18

**Authors:** Zhitao Lyu, Yang Gao

**Affiliations:** Department of Geomatics Engineering, Schulich School of Engineering, University of Calgary, 2500 University Drive, N.W., Calgary, AB T2N 1N4, Canada; ygao@ucalgary.ca

**Keywords:** GNSS, urban environments, intelligent weight scheme, low-cost receiver, multipath detection

## Abstract

High-precision positioning with low-cost global navigation satellite systems (GNSS) in urban environments remains a significant challenge due to the significant multipath effects, non-line-of-sight (NLOS) errors, as well as poor satellite visibility and geometry. A GNSS system is typically implemented with a least-square (LS) or a Kalman-filter (KF) estimator, and a proper weight scheme is vital for achieving reliable navigation solutions. The traditional weight schemes are based on the signal-in-space ranging errors (SISRE), elevation and C/N0 values, which would be less effective in urban environments since the observation quality cannot be fully manifested by those values. In this paper, we propose a new multi-feature support vector machine (SVM) signal classifier-based weight scheme for GNSS measurements to improve the kinematic GNSS positioning accuracy in urban environments. The proposed new weight scheme is based on the identification of important features in GNSS data in urban environments and intelligent classification of line-of-sight (LOS) and NLOS signals. To validate the performance of the newly proposed weight scheme, we have implemented it into a real-time single-frequency precise point positioning (SFPPP) system. The dynamic vehicle-based tests with a low-cost single-frequency u-blox M8T GNSS receiver demonstrate that the positioning accuracy using the new weight scheme outperforms the traditional C/N0 based weight model by 65.4% and 85.0% in the horizontal and up direction, and most position error spikes at overcrossing and short tunnels can be eliminated by the new weight scheme compared to the traditional method. It also surpasses the built-in satellite-based augmentation systems (SBAS) solutions of the u-blox M8T and is even better than the built-in real-time-kinematic (RTK) solutions of multi-frequency receivers like the u-blox F9P and Trimble BD982.

## 1. Introduction

High-precision positioning with low-cost global navigation satellite systems (GNSS) in urban environments remains a significant challenge. The industry demand, however, is high for many emerging applications, such as autonomous vehicles and intelligent transportation systems. Although accurate and reliable solutions have been demonstrated in open sky environments with low-cost GNSS receivers, the positioning accuracy will be greatly degraded in urban environments due to significant multipath effects, non-line-of-sight (NLOS) errors, as well as poor satellite visibility and geometry caused by severe signal blockages [1]. In urban environments, the NLOS signal errors for instance, could be unbounded to become as large as hundreds of meters in some severe circumstances. The effective detection of NLOS signals and subsequent elimination and compensation of NLOS signal effects can significantly improve positioning accuracy in urban environments.

Many methods have been proposed for that purpose [2]. The existing methods could be divided into environmental feature aided approach and GNSS self-maintained approach which differ in the type of information used for NLOS signal detection. For environmental features aided approach, the 3D map aided (3DMA) method is popular, including the ranging based 3DMA method and the shadow matching method [3]. The ranging based 3DMA method utilizes the 3D map to perform ray tracing to simulate the signal transmitting path among buildings and trees, and is thus able to determine the visibility of the signals and even correct the multipath effects and NLOS errors [4,5,6,7,8]. The shadow matching method has been designed for dense urban GNSS positioning which utilizes the visibility matching to determine the possible candidate location of the receiver, and so far it is the only method that utilizes signals that are not even being tracked [9,10]. The omnidirectional camera aided method is also widely used which captures the surrounding infrastructures with a wide-angle of view lens (180 degrees) so that the segmented sky area in the image could be used to predict visible satellites after projecting the satellite positions onto the image [2,11].

For GNSS self-maintained methods, they reply on only data from GNSS receivers and therefore reduce the complexity and the cost of the navigation system when compared to the environment feature aided approach. A dual-polarized antenna, for instance, can be applied to detect reflected signals since the right-hand circular polarized (RHCP) signal will be transferred to the left-hand circular polarized (LHCP) signal when reflected [12]. Consistency checking using receiver autonomous integrity monitoring (RAIM) algorithms is another widely used method for NLOS signals detection [13,14], but it is effective only when the majority of the received signals are line-of-sight (LOS) signals, which could not be guaranteed in urban environments where there are many reflectors and obstructions around. Machine learning methods were recently applied to explore the diverse features of GNSS data. Yozevitch et al. (2016), for instance, analyzed the relationship among signal visibility, C/N0, elevation, the 2nd order derivative of pseudorange, and used a decision tree to classify the LOS/NLOS signals. They demonstrated a 77.6% accuracy for the LOS signals and 87.2% for the NLOS signals [15]. Hsu et al. (2017) used a support vector machine (SVM) to classify the LOS/NLOS signals based on features of C/N0, delta C/N0, pseudorange, delta pseudorange, and positioning residual. The obtainable accuracy is 75.4% [16]. An SVM classifier was also applied using the correlation and tracking information inside a software-defined GNSS receiver as the input features, demonstrating an overall classification accuracy of 82.8% in the urban environments [3]. The visibility labels used in those works are generated by the 3D model of near buildings, which however is not very precise because the 3D models are unable to represent the full shape of the infrastructures around. Deep-learning were also proposed to improve the classification accuracy [17], but they come with a much higher computational load. To date, all works consider the signal classification and positioning test only in a static mode and the performance of the classification accuracy and positioning accuracy in the kinematic scene are not validated.

In this paper, we propose a new multi-feature SVM signal classifier-based weight scheme for GNSS measurements to improve the kinematic GNSS positioning accuracy in urban environments. A GNSS system is typically implemented with a least-square (LS) or a Kalman-filter (KF) estimator, and a proper weight scheme is vital for achieving reliable navigation solutions. Many weight schemes have been proposed. The signal-in-space ranging errors (SISRE) considering the signal noises and the satellite orbit errors is one of them [18,19]. This method, however, works only in open-sky environments since no factor of the transmission path in GNSS denied environments has been considered. The two most popular factors to consider when determining the observation weight are the elevation angle [20,21], and C/N0 [22,23,24]. The elevation angle based weight model assumes that the multipath error, atmosphere and other unmodeled site-specific error will increase at lower elevation angles [25], which, however, also works well only in open-sky environments. C/N0 represents the ratio of the carrier power and the noise power of the received signal, which is a good indicator for the quality of the observations in different environments, and a combination of C/N0 and elevation information is often used to improve the weight model [26,27]. However, as C/N0 is very likely to be affected by the multipath effect [28,29], a multipath affected observation is not necessarily indicated with a large gross error. This demonstrates that the observation quality cannot be fully manifested by C/N0 and elevation values. The new weight scheme is based on the identification of important features in GNSS data in urban environments and intelligent classification of LOS/NLOS signals using the support vector machine (SVM) algorithm. With advantage of better interpreting the quality of the GNSS observations with identified features by the SVM classifier, the proposed weight scheme is superior to the traditional weight scheme as it can better model the GNSS measurement NLOS error in urban environments. To validate the performance of the newly proposed weight scheme, we have tested its computational load and successfully implemented it in a real-time single-frequency precise point positioning (SFPPP) system. The dynamic vehicle-based tests with a low-cost single-frequency u-blox M8T GNSS receiver demonstrate that the positioning accuracy using the new weight scheme outperforms the traditional C/N0 based weight model by 65.4% and 85.0% in the horizontal and up direction, and most position error spikes at overcrossings and short tunnels can be eliminated by the new weight scheme compared to the traditional method. It also surpasses the built-in SBAS solutions of the u-blox M8T and is even better than the built-in real-time-kinematic (RTK) solutions of multi-frequency receivers like the u-blox F9P and Trimble BD982.

The remaining of the paper is organized as follows. Firstly, the related existing works are briefly reviewed. Secondly, the methodology for the new weight scheme-based positioning system development, including an improved SVM classifier and a real-time single-frequency precise point positioning (SFPPP) system aided by the new weight scheme, is presented. Thirdly, the experiment setup, LOS/NLOS signal classification, and positioning results along with analysis are provided. Finally, the conclusions and recommendations for future works are given.

## 2. Methodology

The methodology of the SVM based weight scheme consists of two major components: SVM-based GNSS signal classifier and SVM-based weight scheme. They will be described in this section.

### 2.1. SVM Based GNSS Signal Classifier

#### 2.1.1. Feature Selection

The support vector machine (SVM) was firstly used for GNSS signal classification by Xu and et al. [3], and a test that utilizes more features was later conducted by Lyu and Gao [30]. Since the complex information of signal traveling routes in urban environments cannot be fully manifested in C/N0 and elevations, other features must be investigated to identify more information from GNSS observations. In [30], we have analyzed six features for GNSS signal for their correlations with visibility status, which include differenced C/N0, time single-difference ambiguity, time double-difference phase, time double-difference pseudorange, phase consistency, and pseudorange consistency. In this section, the six features of the same configuration will be used for signal classification. The elevation is a widely used feature for signal classification since satellites at lower elevations are more likely to be obstructed by buildings [31]. However, considering that the correlation of visibility to elevation is not essential, thus in this research, the elevation is not used to avoid the introduction of misleading information to the classifiers.

#### 2.1.2. Support Vector Machine Based Signal Classification

A signal classifier could map the feature data into the probabilities of individual classes. There are many classifiers proposed in the literature, among which the support vector machine (SVM) is a popular one that has been adopted in many fields. When equipped with the radial basis function (RBF) kernel, the SVM can apply a non-linear margin for high dimensional features classification. The structure of the RBF SVM classifier used in this paper is given in Figure 1. The input of the classifier is the six features calculated from the GNSS observations, as presented earlier, where P and L represent pseudorange and phase observations respectively, the P consistency and L consistency are calculated from the discrepancy between the observed observations and the Doppler predicted observations. The output of the classifier is the probabilities of the input observation belongs to LOS (*P_LOS_*) or NLOS (*P_NLOS_*). The open-source software libsvm [32] is used for training and testing in this work.

### 2.2. The SVM Based Weight Scheme

A new weight scheme based on the SVM classifier is developed in this section. The new weight scheme could utilize the intelligent identification of important features in GNSS data in urban environments and intelligent classification of LOS/NLOS signals from the SVM based classifier. The probabilities of the signal being NLOS or LOS are a good indicator of the observation quality. In order to integrate this indicator into the GNSS estimator, a weight scheme is required to map it into observation error covariance. It was indicated that the real-world unmolded GNSS observation error could be modeled as the combination of LOS error and NLOS error [26]. Thus, weight schemes will be considered as the sum of the LOS part and the NLOS part for Doppler, phase, and pseudorange observations.

#### 2.2.1. Doppler Observation Weight Scheme

For LOS Doppler observation error, the equal weight model is effective as the differences are minor for the observation errors at different elevation angles [33]. Thus, the equal weight model is adopted for the LOS part of Doppler observation error, and the covariance for satellite *j* could be written as:(1)$$C_{d,\;l}^{j}=\sigma_{d,l}^{2}$$
where $\sigma_{d,l}$ indicates the LOS part of Doppler observation error standard deviation, which is taken as 0.06m/s in this paper. When Doppler observation is contaminated by the NLOS effect, the model used for the NLOS error part follows a similar form proposed by Suzuki et al. [34], which could be written as:(2)$$C_{d,n}^{j}=\{\begin{array}{cc} b\cdot \left(e^{\left(\frac{1}{1-P_{NLOS}}\right)}-e\right) & P_{NLOS}<0.95\\ b\cdot \left(e^{\left(\frac{1}{1-0.95}\right)}-e\right) & P_{NLOS}\geq 0.95 \end{array}$$
where *b* is an empirical constant value to be tuned. *e* is the Euler’s number, *e ≈* 2.718, *P_NLOS_* indicates the probability of the observation being NLOS, which is the output of the signal classifier. When *P_NLOS_* is larger than 0.95, the covariance is fixed to the value at *P_NLOS_* = 0.95 to avoid introducing astronomical numbers as it will corrupt the 64-bit float data width-based estimator.

The final covariance for Doppler observation error for satellite *j* could be calculated by adding the LOS part and NLOS part:(3)$$C_{d}^{j}=C_{d,l}^{j}+C_{d,n}^{j}$$

#### 2.2.2. Phase and Pseudorange Observation Weight Scheme

For the LOS part of phase and pseudorange observation, the elevation model is accurate enough and has been widely applied. The simplified covariance for LOS phase/pseudorange observations could be written as [35]:(4)$$C_{P,l}^{j}=\frac{a}{sin^{2}\;\left(ele^{j}\right)}$$
(5)$$C_{L,l}^{j}=d\cdot \frac{a}{sin^{2}\left(ele^{j}\right)}$$
where $C_{P,l}^{j}$ and $C_{L,l}^{j}$ represent the covariance of pseudorange and phase observations LOS error for satellite *j*; *a* is an empirical constant value to be tuned; *ele* indicates the elevation angle in rad. *d* is the scale factor between phase and pseudorange observation error, taken as 0.01 in this work. The C/N0 information is not considered in the LOS part, as it has already been interpreted by the signal classifier, which will be used for the NLOS part. The proposed NLOS part for pseudorange and phase error could be written as:(6)$$C_{P,n}^{j}\;or\;C_{L,n}^{j}=\{\begin{array}{cc} 0 & P_{NLOS}<0.4\\ c\cdot \left(e^{\left(10\cdot P_{NLOS}\right)}-1\right) & 0.4\leq P_{NLOS}<0.95\\ c\cdot (e^{9.5}-1) & 0.95\leq P_{NLOS} \end{array}$$
where $C_{P,n}^{j}$, $C_{L,n}^{j}$ represent the covariance of NLOS pseudorange and phase observation error for satellite *j*; *c* is an empirical constant variable to be tuned and will have different values for pseudorange and phase observations. The threshold for considering NLOS error is *P_NLOS_* = 0.4 instead of 0.5 due to some unavoidable misclassification of the signal classifier. Thus, when the *P_NLOS_* is smaller than 0.4, this observation will be taken as LOS signal. In this model, the covariance will increase exponentially as the *P_NLOS_* is above 0.4 to handle the significant increase of the NLOS error. As in the Doppler weight scheme, when *P_NLOS_* is larger than 0.95, the covariance is fixed to the value at *P_NLOS_* = 0.95 to avoid astronomical numbers from corrupting the estimator.

The final covariance for pseudorange/phase observation error could be calculated by adding the LOS part and NLOS part:(7)$$C_{P}^{j}=C_{P,l}^{j}+C_{P,n}^{j}$$
(8)$$C_{L}^{j}=C_{L,l}^{j}+C_{L,n}^{j}$$
where $C_{P}^{j}$, $C_{L}^{j}$ represent the covariance of pseudorange and phase observation error for satellite *j* respectively.

## 3. Methodology Evaluation

To validate the performance of the newly proposed weight scheme, we have implemented it into a real-time Doppler aided single-frequency precise point positioning (SFPPP) system developed at The University of Calgary [36]. A feature of the SFPPP system is that it employs two separate Kalman filters: one for position determination using pseudorange and phase observations and the other for velocity determination using Doppler observations. The position solutions with this approach are more robust than processing all observations in a single filter when with undetected NLOS errors in urban environments. A consistency check based on a chi-square test is also implemented in the SFPPP system to ensure the integrity of the position solutions. If the test fails, the observation with the largest residual would be eliminated for another iteration of estimation.

Figure 2 shows the architecture of the SVM based weight scheme aided SFPPP system. The raw GNSS observations are first input into the SVM based signal classifier. Then the calculated probability of the signal being NLOS would be passed into the proposed weight scheme to calculate the covariance for Doppler, phase and pseudorange observations. After that, the weighted observations are used by the SFPPP system for position determination.

### 3.1. Field Test Description

Two kinematic vehicle-based field tests were performed with a low-cost GNSS receiver (u-blox M8T) in the downtown of Calgary, on 4 August (field test 1) and 12 October (field test 2) 2020, respectively. Figure 3 and Figure 4 show the routes of the two field tests in Google Earth view, in which the yellow triangles and green triangles represent the short tunnel and the pedestrian overcrossing connecting buildings. Figure 5 gives a demonstration of an overcrossing and a short tunnel. As it is shown, the two testing routes include the most challenging scenes in urban environments with overcrossing, short tunnel and urban canyon populated by high-density buildings. In both of the field tests, the vehicle was moving at a maximum speed of 50 km/h (speed limitation in urban Calgary). The filed test on 4 August lasted about 10 min within a small loop square route and will be used to train and assess the SVM signal classifier. The filed test on 12 October lasted 30 min and will be used for positioning accuracy evaluation. The route of field test 2 covered a wider Calgary downtown area including wide streets of six lanes and narrow streets of only two lanes, and the short tunnels and overcrossing over the testing route made the environment further challenging for GNSS positioning.

For GNSS data acquisition, an elevation cutoff angle of 5 degrees was adopted with a data sampling rate of 10 Hz and GNSS data from three constellations (GPS, GLONASS and Galileo) were logged. For the real-time SFPPP system, the orbit, satellite clock bias and real-time ionosphere products from CNES are used for direct corrections [37]. The tropospheric error is corrected using the Saastamonion model with the Global Mapping Function (GMF) [38] for both the zenith wet part and the hydrostatic part. The receiver and satellite phase biases $b_{L1}^{r}$ and $b_{L1}^{s}$ are absorbed by the ambiguity term. The code bias $b_{P1}^{s}$ is corrected using the 30 days differential code bias (DCB) product from the Center for Orbit Determination in Europe (CODE). The empirical parameters for the weight scheme are tuned using the dataset from field test 1, and the used values are given in Table 1. Those parameters are set the same for all satellites.

Three GNSS receivers were used which include a single-frequency receiver (u-blox M8T) and two multi-frequency receivers (u-blox F9P, Trimble BD982), all connected to the same antenna installed on the vehicle roof using a signal splitter. Only raw observations from single-frequency GNSS receiver u-blox M8T will be used in this work. The built-in RTK solutions from the two multi-frequency receivers will be used only for positioning accuracy comparisons.

An upward fisheye camera was set up on the vehicle roof to capture the image of the surrounding environments and thereafter to provide satellite visibility ground truth for the training and testing the SVM model. More details will be provided in the latter of this section. The SPAN system from Novatel, which includes a high-end Novatel Propack6 GNSS receiver and a tactical IMU with a built-in fiber optical gyroscope, was used to provide the reference values (accurate at decimeter-level in this urban environment) for positioning accuracy analysis. Further, the position output from two multi-frequency GNSS receivers (Trimble BD982 and u-blox F9P) were logged for external evaluation of the positioning accuracy of the SFPPP system using the new weight scheme. All GNSS receivers used in the tests were connected to the same GNSS antenna using a signal splitter. A Trimble Net R9 GNSS receiver was set on the roof of ENF building in the University of Calgary to serve as the base station for the Novatel SPAN system, u-blox F9P, and Trimble BD982 receivers, and the maximum baseline length during the whole filed test is about 6.2 km.

### 3.2. SVM Classifier Training and Performance Evaluation

This section focuses on the SVM-based classifier training and performance evaluation. The performance evaluation will be conducted in the testing accuracy aspect and the testing phase computational load aspect to validate the practicality of the SVM based classifier in a real-time GNSS application.

To get the satellite visibility ground-truth for training and testing the SVM based classifier, the upward fisheye camera based satellite visibility labeling method is applied [30]. Firstly, the captured image is segmented into sky areas and obstruction areas manually. Secondly, the satellite location is projected to the image plane via the Mei’s fisheye model [39]. After that, the satellite visibility ground truth could be obtained via comparing the projected satellite location to the segmented image. Figure 6 is an example of this labeling process. An upward fisheye camera image is shown in the left part, and the right part is the segmented image with the projected satellites’ location, in which the segmented sky areas are rendered as blue and the obstruction areas are rendered as black. Those satellites located in the blue areas will be marked as LOS satellites and vice versa.

In the experiment of field test 1, 6078 images were captured using an upward fisheye camera installed on the roof of the vehicle, and they were segmented manually using a labeling tool developed at the University of Calgary to determine the satellite visibility for all GNSS observations. The 70% part of the data in field test 1 is used for training and the rest 30% part is assigned to testing. No data from field test 2 are used for training or testing as the heavy workload of manually labeling.

Table 2 shows the LOS, NLOS and overall accuracy of the trained classifier on the testing dataset. The overall testing accuracy reaches 86.05%, which, however, is much worse than the accuracy from the static test that we presented in [30]. This indicates that the complex information of signal traveling routes in urban environments is harder to interpret according to the six features in the kinematic scene than in the static scene, as the signal traveling route changes much faster.

Table 3 gives out the configuration and result to test the SVM based classifier prediction time. A total number of 1800 epochs of GNSS observations are input sequentially into the trained SVM based classifier to get the satellites visibility prediction. The result reveals that the prediction process for the GNSS observations from a single epoch takes only 11.7 ms. The SVM based classifier consumes a low computational load and can be applied to high-rate real-time GNSS applications, especially given that only one of the four cores of the low-power Intel CPU i5 8250U was used in this test.

### 3.3. Positioning Accuracy Analysis

The evaluation of the proposed new weight scheme will be conducted in two ways. First, the positioning accuracy will be compared between the positioning solutions using the new weight scheme (SVM method) and the widely used C/N0 based weight model (C/N0 method) [22,24]. Then the positioning accuracy will be compared between the positioning solutions using the new weight model and independent position output from commercial receivers including high-end multi-frequency RTK solutions. The trained SVM classifier using the dataset in field test 1 will be directly applied to the dataset in field test 2 to validate the positioning performance. Since no data in field test 2 is involved in the training phase, the positioning performance shown below is expected to be reproducible.

Figure 7 compares the positioning error in time series and the positioning error cumulative distribution function (CDF) of the SFPPP solutions using the SVM method and the C/N0 method. There is a significant gain in terms of position solution robustness for the SVM based weight scheme over the C/N0 based weight scheme. When the SVM method is applied, the number of epochs with positioning error over 10 m in either horizontal or upward directions is greatly decreased compared to the C/N0 method, and the number of position error spikes is also significantly reduced. Further, the overall accuracy of the position solutions is greatly improved by the proposed SVM based weight scheme. The 95% CDF of the positioning errors using the C/N0 method reaches 15 m, 30 m and more than 40 m in the east, north and up directions, while the SVM based weight scheme brings down the 95% CDF of the positioning errors to 5 m, 10 m, and 12 m in the three directions.

It is worth mentioning that the position errors for the SVM method have a mean value close to zero in all three directions, while a noticeable positive bias can be observed in the up direction for the C/N0 method. Such bias is caused by the undetected NLOS errors, as the NLOS effect will always bring positive signal delays to the observations, and the NLOS error would typically grow higher at lower elevations. This further demonstrates the benefit of the newly proposed weight scheme in the detection and proper handling of the NLOS error over the C/N0 method in urban environments.

Table 4 shows the comparison of the statistics for the standard deviation (STD) and the root mean square (RMS) of positioning accuracy for the two methods. The RMS position errors of the SVM method are 7.8 m horizontally and 5.8 m vertically, which are 65.4% and 85.0% improvement in the horizontal and up directions respectively when compared to the traditional C/N0 method.

Figure 8 shows the comparison of the Google earth view of the track of the SFPPP using the C/N0 weight scheme and the SVM weight scheme. It can be seen that, there was position error spikes with the C/N0 method at almost every overcrossing and short tunnel, and the errors can be up to several blocks in some severe situations. As a comparison, the position solutions with the SVM method could stay tightly with the ground truth most of the time, even with an overcrossing or a short tunnel.

In Figure 7, we notice that there is a significant position error spike at the epoch 550s in the east direction and two minor position error spikes at the epoch 180s and 1120s in the east and north direction for the SVM method. They can also be observed in Figure 8 at three locations, and it takes some epochs before the SFPPP position solution converges close to the ground truth after the position error spike occurs. This happens due to the misclassification of the SVM based signal classifier. This issue could be addressed through integration with a low-cost inertial measurement unit (IMU) to provide more information for NLOS detection and position aiding. This will be considered in future work.

To further validate the benefit of the SVM based weight scheme, the SFPPP positioning accuracy with u-blox M8T using the SVM based weight scheme is also compared to the built-in solutions from some commercial receivers, namely the built-in SBAS solutions of the single-frequency u-blox M8T, the built-in RTK solutions of a dual-frequency u-blox F9P, and the built-in RTK solutions of multi-frequency Trimble BD982. The result is given in Table 5. First, it can be seen that, in urban environments, the receiver output RTK solutions from multi-frequency GNSS receivers like the u-blox F9P and the high-end Trimble BD982 have fix rates of only 16.8% and 27.7%, respectively. Second, the positioning accuracy of the proposed method using real-time SFPPP with the u-blox M8T is better than the built-in SBAS solutions of the u-blox M8T, with an RMS improvement of 49.5% and 83.7% in the horizontal and up directions, and further it outperforms the built-in RTK solution from multi-frequency GNSS receivers like the u-blox F9P and the Trimble BD982. The comparison to independent receiver position output further confirms the effectiveness of the proposed new weight scheme for precise positioning in urban environments. It is expected that the new weight scheme can further improve the positioning accuracy in urban environments using multi-frequency receivers and multi-frequency precise point positioning (PPP) and RTK methods.

## 4. Conclusions and Recommendations

In this paper, a new SVM signal classifier-based weight scheme for GNSS measurements has been proposed to improve the kinematic GNSS positioning accuracy in urban environments. Traditionally C/N0 and elevation angle are widely considered for the weight of GNSS measurements, however, they cannot fully manifest the observation quality in urban environments. The new weight scheme is based on the identification of important features in GNSS data in urban environments and intelligent classification of LOS/NLOS signals. With advantage of better interpreting the quality of the GNSS observations with identified features by the SVM classifier, the proposed weight scheme is superior to the traditional weight scheme as it can better model the GNSS measurement NLOS error in urban environments.

The new weight scheme has been tested for its computational load and successfully implemented into a real-time single-frequency precise point positioning system to validate its performance. The dynamic vehicle-based tests with a low-cost single-frequency u-blox M8T GNSS receiver demonstrate that the positioning accuracy using the new weight scheme outperforms the traditional C/N0 based weight model by 65.4% and 85.0% in the horizontal and up direction, and most position error spikes at overcrossing and short tunnels can be eliminated by the new weight scheme compared to the traditional method. It also surpasses the built-in SBAS solutions of the u-blox M8T and is even better than the built-in RTK solutions of multi-frequency receivers like the u-blox F9P and Trimble BD982.

For future work, a low-cost IMU will be integrated with the GNSS solutions to refine the few major positioning error spikes shown in the test. Also, more feature data with other combinations from multi-frequency receivers will be used to increase the classification accuracy and thereafter further improve the accuracy and robustness of the positioning system in urban environments.

## Figures and Tables

**Figure 1 sensors-20-07265-f001:**
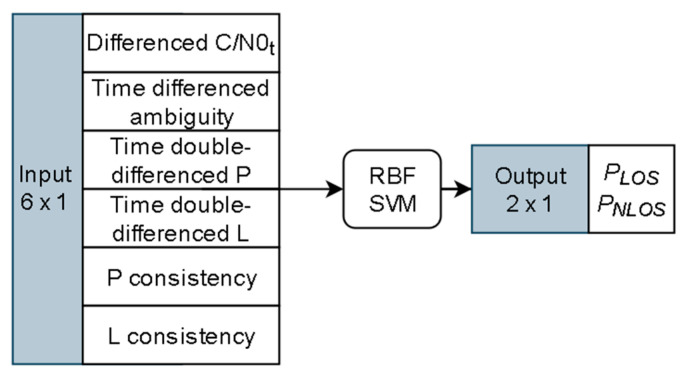
Structure of SVM signal classifier.

**Figure 2 sensors-20-07265-f002:**
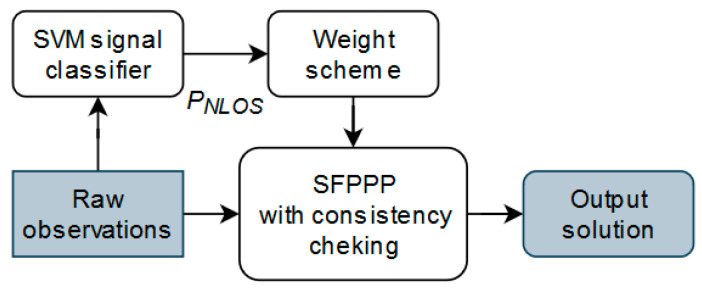
The architecture of the SVM based weight scheme based SFPPP system.

**Figure 3 sensors-20-07265-f003:**
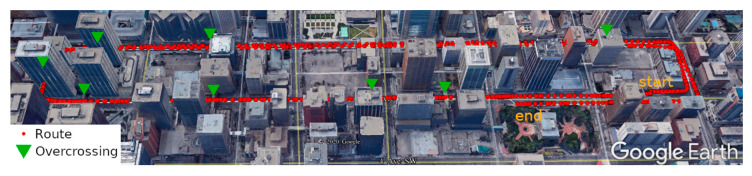
Google Earth view of field test 1.

**Figure 4 sensors-20-07265-f004:**
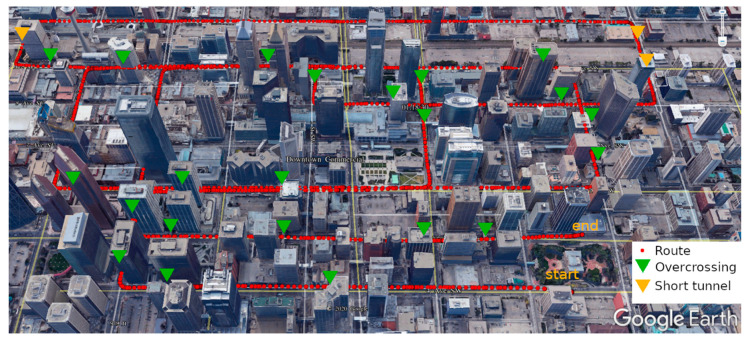
Google Earth view of field test 2.

**Figure 5 sensors-20-07265-f005:**
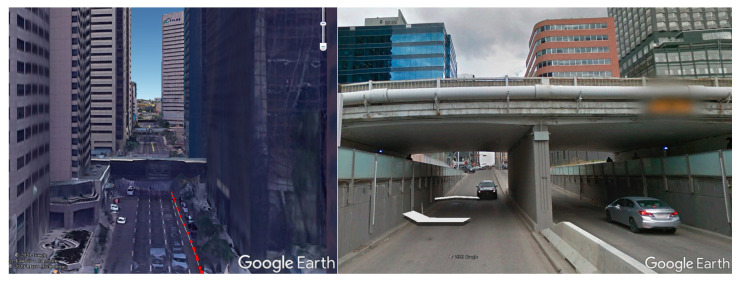
Demonstration of the overcrossing (**left**) and short tunnel (**right**) in Google Earth View.

**Figure 6 sensors-20-07265-f006:**
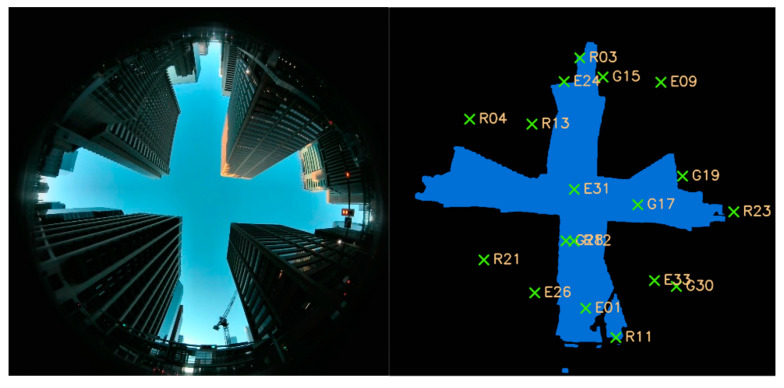
Demonstration of upwards fisheye camera based signal visibility ground-truth determination.

**Figure 7 sensors-20-07265-f007:**
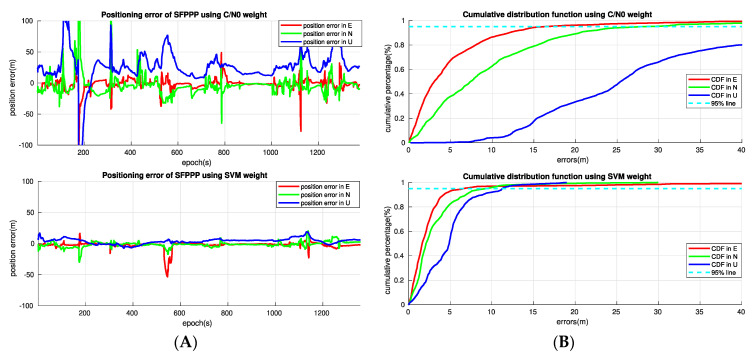
SFPPP positioning error in time series (**A**), and positioning error cumulative distribution functions (**B**) of the two methods.

**Figure 8 sensors-20-07265-f008:**
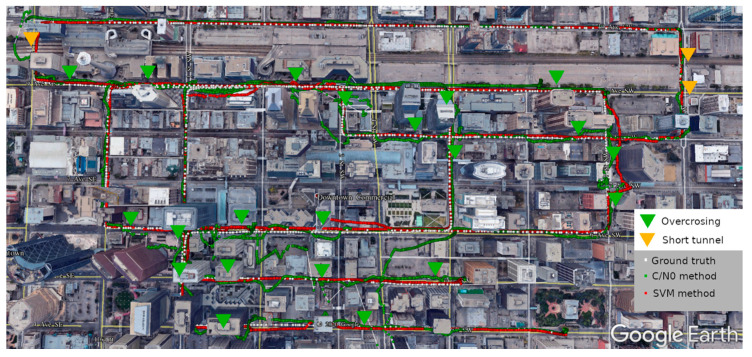
Google Earth view of the C/N0 weight scheme and SVM weight scheme based SFPPP track.

**Table 1 sensors-20-07265-t001:** Parameter values used in the weight scheme.

Parameter	Value	Parameter	Value
$\sigma_{d,\;L}$	0.06 m/s	*c* (for phase)	1 m^2^
*b*	0.2 m^2^/s^2^	*c* (for pseudorange)	5 m^2^
*a*	1 m^2^	*d*	100

**Table 2 sensors-20-07265-t002:** The LOS/NLOS/Overall accuracy for the training dataset and testing dataset.

Accuracy Type	Testing Accuracy
LOS	93.37%
NLOS	81.48%
Overall	86.05%

**Table 3 sensors-20-07265-t003:** Configuration and results to test the SVM based classifier prediction time.

CPU	Intel i5 8250U (3.4ghz boost, single core used)
Data length	3 min, 29,852 observations, 1800 epochs
Data rate	10 Hz
Processing time	21 s
Average time per epoch	11.7 ms

**Table 4 sensors-20-07265-t004:** Positioning accuracy improvement compared to traditional C/N0 weight model.

Method	Direction	STD (m)	RMS (m)	RMS Improvement
C/N0 method	Horizontal	22.374	22.708	
	Up	28.688	38.596	
SVM method	Horizontal	7.391	7.858	65.4%
	Up	4.303	5.804	85.0%

**Table 5 sensors-20-07265-t005:** Positioning accuracy improvement compared to the commercial receiver output.

Receiver	Solutions Type	Direction	STD (m)	RMS (m)	RMS Improvement
u-blox M8T (low-cost single-frequency receiver)	SBAS solutions	Horizontal	15.566	15.574	49.5%
		Up	27.523	35.566	83.7%
u-blox F9P (Low-cost dual-frequency receiver)	RTK solutions (Fix rate: 16.8%)	Horizontal	8.852	8.896	11.7%
		Up	15.663	19.554	70.3%
Trimble BD982 (High-end multi-frequency receiver)	RTK solutions (Fix rate: 27.7%)	Horizontal	12.275	12.275	36.0%
		Up	12.331	13.238	56.2%

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
