# Peer review of "An SVM Based Weight Scheme for Improving Kinematic GNSS Positioning Accuracy with Low-Cost GNSS Receiver in Urban Environments"

_sensors, 2020, doi:10.3390/s20247265_

Round 1
Reviewer 1 Report
I have read the paper thoroughly and it might be a good work based on testing.
Here are some of my comments.
- The idea is an improved version of the scheme which basically doesn't reflect any novelty.
- Why the noise has not been considered while simulating the results.
- What kind of sensors? CDF also don't reflect any high precision and accuracy
- weight scheme is also not novel, as in localization WCL methods are well known which introduce weight for the first time
Reviewer 2 Report
In the paper entitled “An SVM Based Weight Scheme for Improving Kinematic GNSS Positioning Accuracy with Low-cost GNSS Receiver in Urban Environments” the authors propose a new multi-feature support vector machine (SVM) signal classifier based weight scheme for GNSS measurements to improve the kinematic GNSS positioning accuracy in urban environments.
The idea of the research is interesting and presents some novelty.
The results are very interesting and useful in the field of structural monitoring.
The manuscript topics fit enough to the journal scope.
A large number of previous researches by the authors and others have been discussed and those results have been compared to the results of the current research.
The conclusion explains what the current study found and should talk about what future studies should accomplish. Other main general findings are probably buried in the main section of the paper. I invite the authors to improve the paper conclusion in this sense.
Reviewer 3 Report
Dear Editor,
my comments for Authors:
- In Abstract please add the information about your findings and obtainted results.
- all acronyms must be explained in text, please check it.
- Table 1. How you designated the a priori value of these parameters? Are these value a empirical constant? Please explain how you calculate these values? Are these values the same in GPS, GLONASS and Galileo system?
- Figure 7 is illegible, please correct it.
- In paragraph 3.1 you read that GPS/GLONASS and Galileo data will be used in computations. However in the rest text I don't see the reference to this solution. The results from Figure 7, tabel 4 and 5 are estimated using GPS/GLONASS/Galileo data?
- conclusion must included also the obtained results from research test. The results must underline why your research method is better.
Round 2
Reviewer 1 Report
n/a
Reviewer 3 Report
Dear Editor,
I accept the paper in current version.